

# Stem cell imaging through convolutional neural networks: current issues and future directions in artificial intelligence technology

Ramanaesh Rao Ramakrishna[1], Zariyantey Abd Hamid[1], Wan Mimi Diyana Wan Zaki[2], Aqilah Baseri Huddin[2] and Ramya Mathialagan[1]

[1] Biomedical Science Programme and Centre for Diagnostic, Therapeutic and Investigative Science, Faculty of Health Sciences, Universiti Kebangsaan Malaysia, Kuala Lumpur, Malaysia
[2] Department of Electrical, Electronic & Systems Engineering, Faculty of Engineering & Built Environment, Universiti Kebangsaan Malaysia, Bangi, Selangor, Malaysia

## ABSTRACT

Stem cells are primitive and precursor cells with the potential to reproduce into diverse mature and functional cell types in the body throughout the developmental stages of life. Their remarkable potential has led to numerous medical discoveries and breakthroughs in science. As a result, stem cell–based therapy has emerged as a new subspecialty in medicine. One promising stem cell being investigated is the induced pluripotent stem cell (iPSC), which is obtained by genetically reprogramming mature cells to convert them into embryonic-like stem cells. These iPSCs are used to study the onset of disease, drug development, and medical therapies. However, functional studies on iPSCs involve the analysis of iPSC-derived colonies through manual identification, which is time-consuming, error-prone, and training-dependent. Thus, an automated instrument for the analysis of iPSC colonies is needed. Recently, artificial intelligence (AI) has emerged as a novel technology to tackle this challenge. In particular, deep learning, a subfield of AI, offers an automated platform for analyzing iPSC colonies and other colony-forming stem cells. Deep learning rectifies data features using a convolutional neural network (CNN), a type of multi-layered neural network that can play an innovative role in image recognition. CNNs are able to distinguish cells with high accuracy based on morphologic and textural changes. Therefore, CNNs have the potential to create a future field of deep learning tasks aimed at solving various challenges in stem cell studies. This review discusses the progress and future of CNNs in stem cell imaging for therapy and research.

# INTRODUCTION

Stem cells represent a new frontier for regenerative medicine and various therapeutic applications. Stem cells are unspecialized cells in the human body. They have the ability

Corresponding author
Zariyantey Abd Hamid,
zyantey@ukm.edu.my

to undergo self-renewal division and create more stem cells that are fundamental to the maintenance of an undifferentiated stem cell pool. Specifically, self-renewal division allows a stem cell to produce two daughter stem cells of different cellular fates, with one cell undergoing terminal differentiation while the other one maintains its self-renewal properties (*Wilson, Laurenti & Trumpp, 2009*). Stem cells also have the remarkable potential to initiate differentiation into specialized progenitor cells and subsequently differentiate into terminally differentiated mature and functional cells in respective organs. Stem cell populations are established in niches that are known as "in vivo microenvironments," where the stem cells can reside and receive stimuli that regulate their cellular fate (*Zhang & Li, 2008*).

In general, stem cells are commonly classified into embryonic stem cells (ESCs), adult stem cells (ASCs), and induced pluripotent stem cells (iPSCs) (*Alvarez et al., 2012*). ESCs are pluripotent stem cells that can differentiate into all derivatives of the ectoderm, endoderm, and mesoderm germ cell layers that consist of lineage-specific stem cells, such as hematopoietic stem cells (HSCs), mesenchymal stem cells (MSCs), and neural stem cells (NSCs). Thus, the germ layers contain stem cells that are crucial to the development of various tissue types in the body and which are currently being used to treat various pathological disorders (*Mimeault, Hauke & Batra, 2007*). In contrast, ASCs are multipotent stem cells that have less differentiation potency than ESCs. Of note, ASCs are tissue-specific stem cells that can only undergo differentiation in tissue-specific cells, such as blood cells, skin, bone, cartilage, and cardiac muscles (*Pessina & Gribaldo, 2006*). Lastly, iPSCs are obtained by genetically reprogramming mature cells into embryonic-like stem cells. Via iPSC technology, terminally differentiated cells are induced into becoming pluripotent again, at which point they can function similarly to ESCs (*Yamanaka & Blau, 2010*).

Recently, iPSCs have gained popularity because their usage does not raise major ethical issues, as ESCs do (*Fan et al., 2017*). Moreover, iPSCs are known to express specific genes and proteins as well as differentiation patterns that are believed to be identical to ESCs (*Chagastelles & Nardi, 2011*; *Nordin, Ahmad & Farzaneh, 2017*). In addition, iPSCs are known to assist in detecting the earliest disease-causing events in cells, making them extremely beneficial in various cell-based therapies (*Kavitha et al., 2017*). However, there is a limitation associated with using iPSCs in medicine. Human error is possible while evaluating iPSCs quality through colony morphology, which is a crucial obstacle to overcome prior to attempting further experimental or therapeutic approaches. This problem calls for an automated system that can reduce errors and provide optimal iPSC analysis. To overcome this limitation, the application of artificial intelligence (AI) technology is recommended. Using AI with deep learning–based algorithmic frameworks has become a common approach to solving various problems related to iPSC analysis (*Kusumoto & Yuasa, 2019*). The major fields for AI include machine learning (ML), natural language processing (NLP), computer vision, robotics, and autonomous vehicles. ML is an algorithm that allows a computer to learn the formation and classification of patterns in large datasets without being explicitly programmed to do so. One form of ML is deep learning, which learns data features using a multi-layered neural network that mimics the human neural circuit structure (*Kusumoto & Yuasa, 2019*).
The convolutional neural network (CNN), which is one type of supervised approach to deep learning, has improved the results of image recognition studies (*Krizhevsky, Sutskever & Hinton, 2012*; *Zeiler & Fergus, 2014*; *Szegedy et al., 2015*; *He et al., 2015*; *Zeng et al., 2016*). However, despite the findings reported in these studies about the potential applications of AI in medical image analyses, such as for breast cancer detection (*Azli et al., 2018*; *Rodríguez-Ruiz et al., 2019*) and lung pathology (*Van Riel et al., 2017*; *Hwang et al., 2019*), studies into the applications of AI in stem cell analysis have been limited. Therefore, we believe that AI-based technology, specifically CNNs, can have a great impact on stem cell biology research, as the performance of CNNs has been proven to equal the performance of human experts (*Rawat & Wang, 2017*; *Yadav & Jadhav, 2019*).

To date, only a few of studies concerning the use of AI in medical fields were focusing on the use of CNNs in stem cell studies, specifically for iPSCs. These reports describe the potential applications of CNNs in the recognition of cell regions of human iPSCs using microscopy images (*Yuan-Hsiang Chang et al., 2017*), the recognition and classification of iPSC colonies from microscopy images (*Fan et al., 2017*), the identification of iPSC-derived endothelial cells (iPSC-ECs) without the need for immunostaining and lineage tracing (*Kusumoto et al., 2018*), and the identification of undifferentiated pluripotent stem cells from early differentiating cells with more than 99% accuracy (*Waisman et al., 2019*). Therefore, the prime objective of this review is to highlight and discuss the progress and future of AI-based technology, specifically CNNs, which can be used in the biomedical field, particularly for stem cell–based therapy and research. Moreover, particular attention is given to ongoing studies using AI-based technology as an automated and robust platform for improving the analysis of iPSC-derived colonies and potentially for analyzing other colony-forming stem cells, such as HSCs. This research is critical, as manual analysis of colony-derived stem cells is time-consuming, error-prone, and training-dependent. Thus, this review is intended for research scientists, medical practitioners, educators, and students interested in using AI-based technology, particularly CNNs, for biomedical and stem cell biology applications.

This review is organized as follows. "Survey Methodology" elaborates on the search criteria we used to find articles and references. "Overview of Stem Cells" discusses the properties of stem cells. "Stem Cell–Based Therapy" discusses the use of stem cells in treatments of diseases. "Emerging Artificial Intelligence Technology" discusses the rise of AI and how it is employed as a novel technology in various sectors. "Machine Learning" provides information on the relationships between ML and AI, the types of ML, and how ML works. "Deep Learning" discusses the relationship of DL to ML and how DL works. Current applications of CNNs, the history of CNNs, and a discussion of how CNNs learn are provided in "Convolutional Neural Networks." The applications of CNNs in medical analysis and cell biology are discussed in "Convolutional Neural Networks in Medical Analysis" and "Convolutional Neural Networks in Cytobiology," respectively. "Convolutional Neural Networks in Pluripotent Stem Cell Studies" reviews the ability of CNNs to identify and classify pluripotent stem cell colonies. Future applications of AI-based CNNs in improving the analysis of other colony-forming stem cells, particularly HSC-derived colonies from myeloid, erythroid, and lymphoid lineages, are described in

"Future Applications of Artificial Intelligence Technology in Hematopoeitic Stem Cell Analysis." Finally, the conclusions of this study are presented in "Conclusions."

## SURVEY METHODOLOGY

Table 1 summarizes the article selection criteria used to construct this review. Briefly, journal databases, mainly Google Scholar and PubMed, were used to research the scholarly articles reviewed in this paper. The keywords used to search for these articles included "AI," "ML," "deep learning," "convolutional neural network," "induced pluripotent stem cell," "hematopoietic stem cell," "biomedical imaging," "medical analysis," "cell biology," "morphology," and "pattern recognition." The inclusion criteria for the selected articles required the articles to be related directly to AI and its technology, medical and biological imaging, and stem cell studies. The searches were not refined by publishing date, authors, author affiliations, journals, or the impact factors of the journals. The quantitative articles provided measurable data that uncovered trends and patterns in medical image analysis via AI. The qualitative articles provided insights into the problems and ideas or hypotheses underlying the applications of AI in various medical sectors. In summary, this review is based on 124 references comprised of 54 original research articles, 48 review articles, five conference papers, five books, four commentary papers, four reports, two webpages, one regular article, and one essay. These references focused on the use of AI technology in medical analysis, cell biology, and stem cell analysis, primarily for colony-forming stem cells.

### Overview of stem cells

Stem cells are defined as undifferentiated cells with the ability to perform self-renewal division and differentiation into any specific cell types of an organism. Self-renewal is a fundamental property of stem cells that distinguishes them from terminally differentiated cells, which allow the stem cells to go through numerous cycles of cell growth and cell division while sustaining their undifferentiated state and tissue regenerative potential (*Molofsky, Pardal & Morrison, 2004*; *Menon et al., 2016*). Stem cells undergo self-renewal division via two distinct models that are crucial for the maintenance of tissue hemostasis (Fig. 1). In the symmetrical division model, a stem cell can divide into two new stem cells while maintaining its self-renewal properties or to two differentiated daughter cells. Alternatively, asymmetrical cell divisions can take place, through which a stem cell gives rise to an identical stem cell, and the other daughter cell undergoes differentiation into a more specialized cell (*Shahriyari & Komarova, 2013*; *Łos, Skubis & Ghavami, 2018*).

Stem cells can be further classified based on their differentiation potency and origin. In terms of potency, stem cells are divided into five main categories: totipotent, pluripotent, multipotent, oligopotent, and unipotent (*Kmiecik et al., 2013*; *Maleki et al., 2014*). Totipotent stem cells are cells that can divide and differentiate into all cell lineages of the whole organism in all three germ layers (mesoderm, endoderm, and ectoderm) as well as into placental cells. These cells are regarded as having the highest differentiation potential, as they are able to generate both embryonic and extra-embryonic cells (*Zakrzewski et al., 2019*). An example of a totipotent cell is the zygote, which is formed from the fertilization

**Table 1  Summary of the article selection criteria used to construct this review.**

| No. | Category | Description |
|---|---|---|
| 1 | Journal Databases | 1. Google Scholar |
| | | 2. Pubmed |
| 2 | Inclusion Criteria | 1. Articles contain the keywords "artificial intelligence", "machine learning", "deep learning," "convolutional neural network," "induced pluripotent stem cell," "hematopoietic stem cell," "biomedical imaging," "medical analysis," "cell biology," "morphology," and "pattern recognition." |
| | | 2. Articles that are related directly to artificial intelligence and its technology, medical and biological imaging, and stem cell studies. |
| | | 3. The searches for articles were not refined by publishing date, authors, author affiliations, journals, or the impact factors of the journals. |
| | | 4. Quantitative articles that provided measurable data that uncovered trends and patterns in medical image analysis via artificial intelligence. |
| | | Qualitative articles that provided insights into the problems and ideas or hypotheses underlying the applications of artificial intelligence in various medical sectors. |
| 3 | Exclusion Criteria | None |
| 4 | Types of articles | 1. 53 original research articles |
| | | 2. 48 review articles |
| | | 3. 5 conference papers |
| | | 4. 5 books |
| | | 5. commentary papers |
| | | 6. 4 reports |
| | | 7. 2 webpages |
| | | 8. 1 regular article |
| | | 9. 1 essay |

of an oocyte by a sperm. Pluripotent stem cells are the cells that can differentiate into all three germ layers, excluding the placenta. Specifically, they can differentiate into any fetal or adult cell types, but not embryos (*Loya, 2014*). ESCs are one example. ESCs are derived from the inner cell mass of a blastocyst. Another example of a pluripotent stem cell is the iPSC, which is artificially produced from somatic cells by induction/genetic reprogramming and displays similar functional properties to the PSC (*Zakrzewski et al., 2019*). Multipotent stem cells can differentiate into discrete cell types. Examples include HSCs, which can develop into all blood cell types (red blood cells, white blood cells, and platelets), and MSCs, which can differentiate into a variety of cell types, including osteoblasts, chondrocytes, and adipocytes (*Mitalipov & Wolf, 2009*). Oligopotent stem cells, which usually are present in a specific tissue, only differentiate into cells of a specific lineage (*Kolios & Moodley, 2013*). Examples include the myeloid and lymphoid progenitor cells, which differentiate into the blood cells of their respective lineages (*Zakrzewski et al., 2019*). Lastly, unipotent stem cells, which have the narrowest differentiation ability,

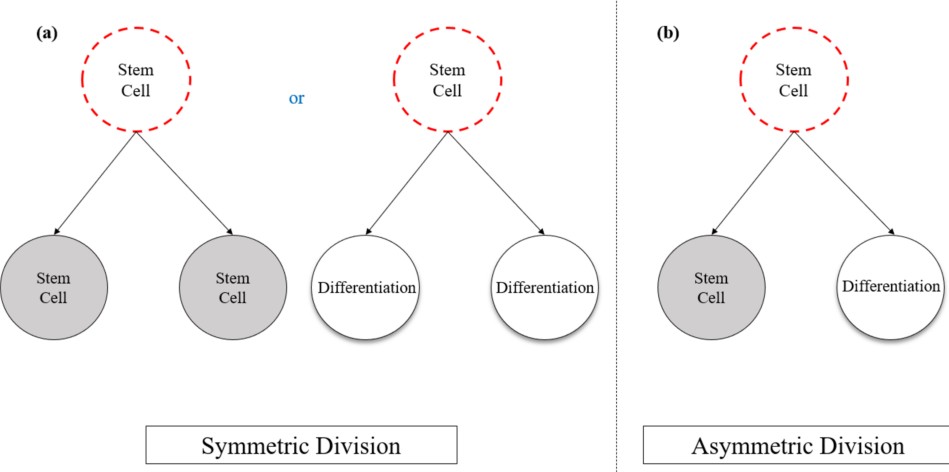

**Figure 1** **Schematic representation of a stem cell division models in relation to self-renewal and the differentiation potential.** (A) In symmetrical division model, a stem cell divides into two new stem cells or two differentiated cells; (B) In the asymmetrical division, one parent stem cell divides into one stem cell and one differentiated cell. Image adapted from *Shahriyari & Komarova (2013)*.

can only produce cells of their own type, though they have a self-renewal property (*Loya, 2014*; *Zakrzewski et al., 2019*). Hepatoblasts, which can differentiate into hepatocytes, are an example of unipotent stem cells.

Stem cells can also be categorized based on their origin. The common categories of stem cells based on their origin are ESCs, ASCs, and iPSCs (*Bongso & Richards, 2004*; *Ilic & Polak, 2011*). Being a pluripotent, ESCs derive from the inner cell mass of the late blastocyst (*Evans & Kaufman, 1981*) and are capable of differentiating into cells in the three germ layers 5–6 days after fertilization (*Łos, Skubis & Ghavami, 2018*). Human ESCs were first obtained in 1998 but were limited in use due to the debatable ethical concerns around their harvesting, which required the destruction of an embryo (*Zhao et al., 2012*; *Wu, 2015*; *Menon et al., 2016*; *Damdimopoulou et al., 2016*). Most ESCs are now generated from eggs fertilized in an in vitro clinic rather than from eggs fertilized in vivo (*Zakrzewski et al., 2019*). ASCs, which are also regarded as somatic stem cells, are multipotent and therefore involved in tissue maintenance and growth (*Seo, Hong & Do, 2017*). They are found in the microenvironment (niche) of the body and among differentiated cells (*Łos, Skubis & Ghavami, 2018*; *Zakrzewski et al., 2019*). Some examples of ASCs include MSCs, which are responsible for generating bone, cartilage, and fat cells; neural cells, which differentiate into nerve cells, oligodendrocytes, and astrocytes; HSCs, which form progenitor and mature blood cells; and skin stem cells, which produce layers of protective skin (*Zakrzewski et al., 2019*). In addition, iPSCs are cells generated from adult somatic stem cells through genetic reprogramming (*Wasik et al., 2014*; *Hirschi, Li & Roy, 2014*; *Hosoya & Czysz, 2016*; *Cieślar-Pobuda et al., 2017*). In 2006, Yamanaka and Takahashi overexpressed four transcription factor genes, encoding Oct4, Sox2, Klf4, and c-Myc into somatic cells, which induced the cells to revert to the pluripotent state (*Takahashi & Yamanaka, 2006*). This breakthrough

made iPSCs a better candidate than ESCs for new drug development, disease modeling, and regenerative medicine (*Kolios & Moodley, 2013*), as the iPSCs exhibited similar self-renewal, morphology, growth kinetics, and gene expression properties to the ESCs without raising major ethical issues (*Łos, Skubis & Ghavami, 2018*).

## Stem cell therapy

The ability of stem cells to build every tissue in the human body makes them a great candidate for therapeutic tissue regeneration. As mentioned above, stem cells are able to undergo self-renewal and differentiation. Self-renewal produces progeny that are identical to the originating cell, whereas differentiation can produce specialized cell types that can repair or replace old or damaged cells (*Reya et al., 2001*). Since the 1960s, multipotent blood stem cells taken from bone marrow have been used to treat blood disorders, such as leukemia, myeloma, and lymphoma, as these cells are able to generate lymphocytes, megakaryocytes, and erythrocytes. Recent studies have investigated the treatment of other diseases using stem cells derived from bone marrow. One such study reported success in engineering an entire articular condyle in rats using MSCs, giving rise to bone and cartilage (*Alhadlaq & Mao, 2005*). In the future, multipotent stem cells are speculated to revolutionize regenerative medicine due to their potentially high plasticity, which may make them capable of differentiating into unrelated cell types.

Another cell with the potential to be used in stem cell therapy is the PSC. Although pluripotent stem cells are likely to be used in stem cell therapy due to their potency, these cells were not used in this field at first, as multiple animal studies resulted in the formation of teratomas, a mixture of cell types from all the germ layers (*Biehl & Russell, 2009*). This issue was solved later when the PSCs were modified to limit their proliferative capacity. This breakthrough allowed many more studies to be conducted on animals to treat a variety of diseases, such as heart failure, muscular dystrophy, and Parkinson's disease (*Laflamme et al., 2007*; *Darabi et al., 2008*; *Wernig et al., 2008*). However, ethical issues still surrounded the use of pluripotent cells in multiple types of therapy. This controversy began because scientists isolating ESCs in the lab had to damage the embryo, which had the potential to become a living organism (*Zakrzewski et al., 2019*). This issue was solved when Yamanaka and Takahashi successfully reverted multipotent ASCs into a pluripotent state (*Takahashi & Yamanaka, 2006*). This new cell, the iPSC, faced no ethical issues, as no embryos were damaged. Following this breakthrough, iPSCs became a promising source of PSCs that could be used in stem cell therapy for the treatment of various diseases.

## Emerging artificial intelligence technology

AI has been used widely in a number of industries related to computerized personal assistants and self-driving cars (*Galbusera, Casaroli & Bassani, 2019*). Recently, AI has also been employed to solve problems and challenges in medical data analysis (*Saad et al., 2019*). AI is an artificial form of human intelligence that can support wide-ranging branches of computer science. AI procedures involve learning, self-thinking, and self-amendment (*Wang, Zhang & Wang, 2019*). In the learning phase, AI machines acquire and standardize data for further use. The output from the learning phase is then used to generate rules in

the thinking phase. Then, during self-amendment, which is the last phase, the system will determine the optimal rules and make changes to the system based on the data obtained.

To date, many AI structures incorporate expert systems, speech recognition, and machine vision. Due to the need in medicine for an automated data analysis system to analyze huge numbers of microscopic images, ML is promising (*Conrad & Gerlich, 2010*; *Lock & Strömblad, 2010*). The main aim of computer data analysis is to maintain objectivity and consistency while processing huge datasets in addition to reducing the workload for researchers (*Danuser, 2011*). Bio-image strategies offer ground-breaking solutions for specific image analysis tasks, such as object recognition, morphometric feature measurements, or motility analysis (*Danuser, 2011*; *Eliceiri et al., 2012*; *Myers, 2012*; *Murphy, 2014*). However, for most biological tests, specific image analysis algorithms have been developed, and using each algorithm for every cell marker and cell type often requires reprogramming the software and adjusting the parameters of which ML offers the most intriguing and promising performance.

## Machine learning

While the terms "artificial intelligence" and "machine learning" are sometimes used interchangeably, ML is actually a branch of AI. The term "machine learning" was introduced in 1959 by Arthur Samuel, who characterized it as the ability of computers to learn without being expressly modified (*Samuel, 2000*). This concept requires humans to furnish machines with the information they require for learning so that the machines can complete tasks or make choices without being programmed to do so. Neural networks, which are defined as a computational learning system to characterize and sort out information like the human brain, have progressed the field of AI. An ML system can make a computed assumption based on the best odds and is even ready to learn from prior errors, making the system "intelligent" (*Adedokun, 2019*).

There are three forms of ML: supervised, unsupervised, and reinforcement learning (*Galbusera, Casaroli & Bassani, 2019*). Supervised learning deals with problems of guideline learning in which the training data includes label samples. The underlying mathematical model will learn its parameters and use the labeled samples to predict and classify the test samples. Unsupervised learning is a form of ML that does not require examples with class labels. Reinforcement learning includes feedback from the environment. Thus, it is not exactly unsupervised. This approach does not possess any label samples for training and consequently cannot be treated as supervised learning either. ML works by studying the processing regulations from model examples instead of depending on manual changes to parameters or processing steps that had been fixed earlier (*Franklin, 2005*; *Bishop, 2006*; *Domingos, 2012*). ML is better than traditional image processing programs due to its ability to solve complex multi-dimensional data evaluation tasks, such as distinguishing between complex morphologies with only a few parameters (*Boland & Murphy, 2002*; *Conrad et al., 2004*; *Neumann et al., 2010*). ML has a long history and is part of many sub-fields, among which deep learning is the primary focus of this study.
## Deep learning

Deep learning is a subset and prevalent field of ML that learns from data via computational modeling of the learning process. It utilizes algorithms to process information, comprehend human speech, and perceive objects visually (*Adedokun, 2019*). Though AI has faced multiple problems over the years with facial recognition, speech recognition, and computer vision, deep learning has provided solutions to all these issues (*Lecun, Bengio & Hinton, 2015*). Artificial neural networks (ANNs) provide the foundation for developing deep learning. An ANN is a common ML technique that simulates the learning mechanism of a biological organism. Inspired by the ability of the human brain to achieve non-linear, parallel, and complex computations, ANNs have been proven to be universal function approximators when given sufficient neurons within the hidden layer of their networks (*Kyaw, Oo & Zaw, 2019*).

Deep learning models ordinarily utilize various hierarchical structures to connect all layers, where the data output from the lower layers will be the input for the higher layers. This characteristic enables deep learning models to change low-level data features into high-level abstract features (*Du et al., 2017*), making the deep learning models better at feature representation than other shallow ML models, such as support vector machines (SVM) (*Cortes & Vapnik, 1995*) and boosting (*Freund & Schapire, 1997*). Figure 2 shows the relationship between AI, ML, ANNs, and DL.

According to Du et al., deep learning has reduced the dependency on users for operation, as this system approach depends on data, in contrast to conventional ML methods that rely on the experience of the user. Along with the progress seen in computer technology, computer performance, and the proliferation of data on the internet, deep learning has developed rapidly and has become an important technique for ML. CNNs are one of the architectural types for deep neural networks and come from a family of multi-layer neural networks principally constructed for two-dimensional data processing, such as images and videos (*Arel, Rose & Karnowski, 2010*).

## Convolutional neural networks

CNNs are one variant of neural networks that are employed primarily in image classification, aggregation based on image similarity, and object recognition. The model for CNNs was inspired and enhanced by the memory processing of the primate visual cortex (*Fu et al., 2016*). Originally, CNNs were extensively applied in object recognition tasks. However, they have now been widely used for object detection and recognition, action recognition, visual labeling, and more (*Jialue Fan et al., 2010*; *Donahue et al., 2013*; *Farabet et al., 2013*; *Jaderberg, Vedaldi & Zisserman, 2014*).

Recently, CNNs have consistently performed well in generic visual recognition tasks (*Krizhevsky, Sutskever & Hinton, 2017*), which has revived a broader interest in CNN-based classification models (*Razavian et al., 2014*). CNNs obtain hierarchical and high-level feature representations by dealing with various levels of input images and are now involved in many computer vision applications, such as auto-driving, robotics, drones, medical diagnostics, and treatments for vision problems. Neocognitron, a self-organizing neural network model for a mechanism of pattern recognition that was built in 1980, has been
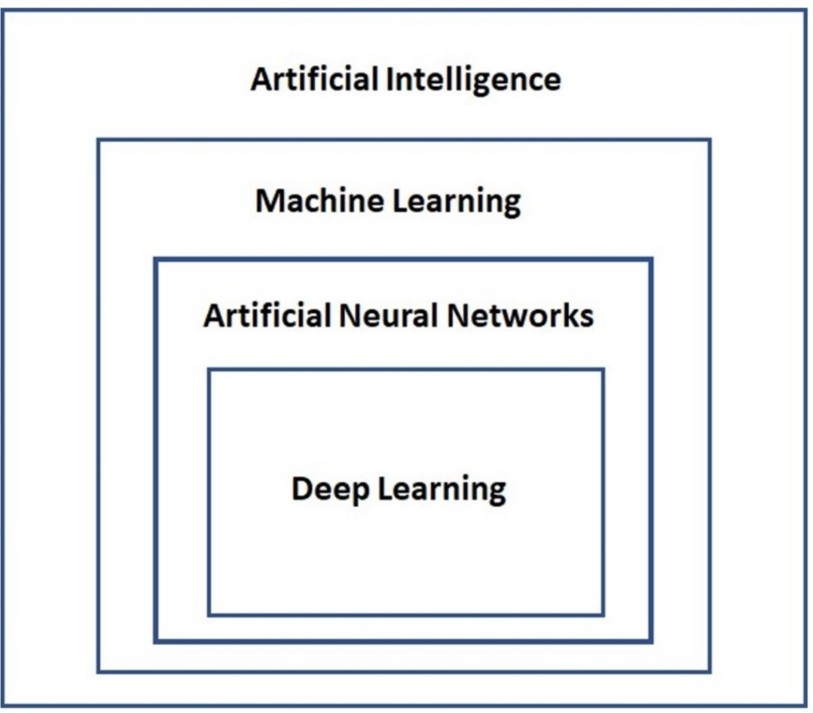

**Figure 2** **Relationship between artificial intelligence, machine learning, artificial neural networks and deep learning.** Artificial Intelligence is the largest scope that contains the subset of Machine Learning. Under Machine Learning is the Artificial Neural Network, and Deep Learning is the subset of Artificial Neural Network.

regarded as the precursor to CNNs (*Fukushima, 1980*). In 1998, LeCun created LeNet, an ANN with multiple layers that became a major contributor to the development of CNNs (*LeCun et al., 1998*). LeNet was constructed to classify handwritten digits and recognize visual patterns from input images with no pre-processing steps. However, this model failed to perform well in dealing with complex problems due to insufficient training data and computing capacity.

In 2012, the ImageNet Large Scale Visual Recognition Challenge (ILSVRC) competition was held, where Krizhevsky and his colleagues produced a CNN model that successfully reduced the annotation errors in large-scale image classification (*Russakovsky et al., 2015*). This model was AlexNet, an in-depth CNN architecture consisting of five layers of convolution followed by three fully connected layers, which brought substantial improvements to image classification tasks. Ever since, AlexNet has been regarded as one of the most prominent developments in the field of computer vision and has been used by many researchers to test variations in the design of CNNs. For example, after AlexNet, several architectures, such as GoogleNet (*Szegedy et al., 2015*), VGGNet (*Simonyan & Zisserman, 2014*), ZFNet (*Zeiler & Fergus, 2014*), and ResNet (*He et al., 2015*), have been developed and have demonstrated boosted performance.
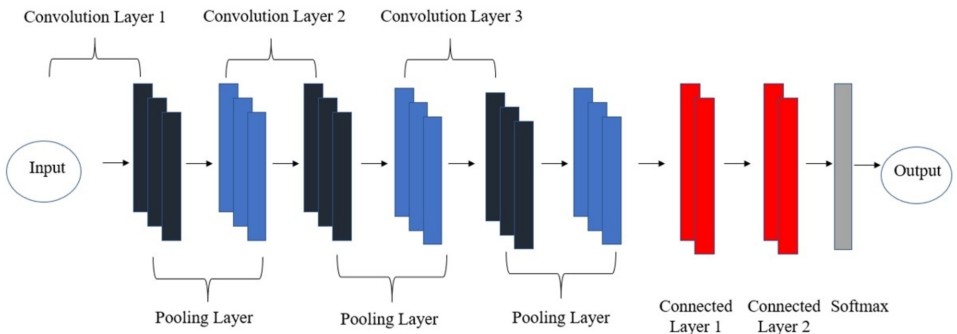

**Figure 3  Convolutional neural network architecture.** The convolutional neural network architecture is composed of convolution layers, pooling layers, connected layers and softmax layer.

CNNs learn image features from basic data. As illustrated in Fig. 3, the CNN architecture is composed of convolution layers, pooling layers, connected layers, and softmax layers. It is still a hierarchical network like the ANN, but the structure and shape of the layers have changed. The CNN framework can be divided into two parts: the feature extraction part (the convolution layers and pooling layers) and the classification part (the fully connected layers). First, an image is passed through a series of convolution layers, then through the pooling layers for feature extraction, and finally through the fully connected layers for classification. The role of the convolution layers is to create a feature map in different sizes, which are then reduced by the pooling layers before being transferred to the upcoming layers. The preliminary layers will identify the simple structures in the image, such as the edges, blobs, and lines, whereas the neurons in the deeper layers will detect the more complex structures (*Anwar et al., 2018*). The fully connected layers produce the required class prediction as an output via probability distribution over the number of $j$ class. A function called "softmax" may be used to predict the output by its probability, $P$, for each $j$ class in a sample vector $\mathbf{x}$:

$$P\left(y=j|\mathbf{x}\right)=\frac{e^{\mathbf{x}^{T}w_{j}}}{\sum_{k=1}^{K}e^{\mathbf{x}^{T}w_{k}}},$$

where $w$ is a weighting vector.

To ensure the CNN performs at a high level, the network must go through a training phase to learn the best possible weights of the images. The CNN gives a better representation of the images as the error signal attained by the loss function is propagated back to improve the feature extraction part. One of the most commonly used optimization algorithms in the training phase for deep learning is the stochastic gradient descent (SGD). The SGD iteratively updates the parameters, such as the weights in the network, by minimizing the cross-entropy loss function, $J(\theta)$, as shown in the following formula:

$$J(\theta)=\frac{1}{j}\sum_{i=1}^{j}H(x_{j},y_{j}),$$

where $\mathbf{x}$ is the desired output class, $\mathbf{y}$ is the actual output class, and $\boldsymbol{H}$ is the cross entropy between $\mathbf{x}$ and $\mathbf{y}$.

## Convolutional neural networks in medical analysis

Medical imaging is a fundamental part of the diagnosis and treatment of illnesses in clinical practices since it produces visual data of the human body. To date, AI is the best-performing technology in healthcare for the analysis of medical images and big data (*Datta, Barua & Das, 2020*). The impact of AI in this field is significant, especially because it assists clinicians in the analysis and interpretation of medical images, has great accuracy, enhances workflow, and reduces medical errors; in addition, it assists patients through the use of algorithms in devices such as smartwatches (*Fingas, 2018*; *Victory, 2018*), recording the patient data and making it available for processing and tracking (*Topol, 2019*). Many recent studies have used AI technologies and their components, particularly ML and DL, to improve healthcare systems (*Dzobo et al., 2020*; *Milstein & Topol, 2020*), support disease and abnormality detection through medical imaging (*Berzin & Topol, 2020*; *Nagendran et al., 2020*; *Thomford et al., 2020*), analyze and handle big data (*Keyes et al., 2020*), and facilitate organ damage detection (*Agur, Daniel & Ginosar, 2002*).

In addition, digital image processing assists in segmentation, classification, and irregularity detection in the analysis of medical images produced by various clinical imaging modalities (*Anwar et al., 2018*). Medical imaging extracts significant data for diagnosis and research purposes, such as the location and divisions of anatomical abnormalities (*Schlegl et al., 2015*) and different body structures (*Rahman, Desai & Bhattacharya, 2008*; *Zaki et al., 2011*). This system also helps clinicians to make diagnoses and prescribe treatments efficiently. Huge datasets of images are generated annually by clinical departments and are assessed by clinical experts, and these images include epidemiological information and markers that are relevant during diagnosis and treatment (*Schlegl et al., 2015*). Due to the growing number of medical images with clinical information, a system is required to handle the big data analysis.

The development of computer vision demonstrates how deep learning methods can be used to manage big data medical image analysis, as evidenced by a recent study where deep learning was applied to medical image analysis groups across the world (*Greenspan, Van Ginneken & Summers, 2016*). CNN is the most suitable model of deep learning for medical imaging, as it excels in learning useful representations of images and data (*LeCun et al., 1998*).

An extensive review showed that CNN networks can provide promising results and can be successful at medical image analysis. The accuracy and performance of the CNN depends on the number of images, the number of classes, and the model of CNN chosen to analyze the images (*Hussain, Anwar & Majid, 2018*). Various studies have proven the success of CNNs in medical image segmentation (*Hussain, Anwar & Majid, 2018*), computer-aided diagnosis (*Pratt et al., 2016*; *Ma et al., 2017*; *Sun et al., 2017*; *Berzin & Topol, 2020*), and disease detection and classification (*Sirinukunwattana et al., 2016*; *Anthimopoulos et al., 2016*; *Van Tulder & De Bruijne, 2016*; *Yan et al., 2016*). As a result, *Anwar et al. (2018)* concluded that CNNs perform better in medical analysis than other methods such as Linear regression and SVM.

## Convolutional neural networks in cytobiology

Microscopes are undeniably important to the medical field and can detect abnormal cells. Detecting cultured cells through a microscope is crucial in cytology, but is tedious when done by a human expert, as this process is time-consuming and error-prone. However, deep learning is capable of tackling this limitation, as it can analyze huge volumes of data efficiently.

According to *Kusumoto & Yuasa (2019)*, molecular biology is another significant field where deep learning has potential, as each cell has unique morphological features. However, few research studies have applied CNNs to cell identification. In silico labeling is a deep neural network introduced by *Christiansen et al. (2018)* that can predict the location and texture of cell nuclei, the state and type of a cell, and the type of subcellular structure using images from bright field microscopy without the use of immunolabeling techniques. For example, every hematopoietic cell from a specific lineage will have unique morphological and cellular properties as a result of its differentiation from the primary stem cells. Thus, deep learning systems use these unique properties to efficiently identify each type of differentiated hematopoietic cell from a microscopy image (*Buggenthin et al., 2017*).

CNNs are effective in identifying human iPSCs with Top-1 and Top-2 error rates of 9.2% and 0.84%, respectively (*Yuan-Hsiang Chang et al., 2017*)). In a research study conducted by *Niioka et al. (2018)*, a CNN was used in the identification of C2C12 cells from phase-contrast images, where the myoblasts were differentiated into myotubes (*Niioka et al., 2018*). The CNN extracted the changes in the morphological features and classified them with 91.3% accuracy. CNN-based semantic segmentation can also classify images at the pixel level by assigning each pixel in an image to a particular object category. In addition, CNNs can allow the detection of object boundaries and object categorization within the defined boundary area. Thus, semantic segmentation is widely practiced in subject areas such as cell biology and the medical sciences (*Kusumoto & Yuasa, 2019*). Semantic segmentation can not only recognize the location of a cell, but also points out the category of the cell as well. Cells can also be sorted without molecular labels through a method called "ghost cytometry," whereby the morphological features are converted to wave information utilizing a barcode system that categorizes and sorts the cells (*Ota et al., 2018*).

## Convolutional neural networks in pluripotent stem cell studies

Pluripotent stem cells are very well known in the fields of regenerative medicine, disease modeling, and drug testing for their ability to become specific to all organism cell types (*Waisman et al., 2019*). ESCs and iPSCs are two forms of pluripotent stem cells (PSCs). The former derives from the early stage of an embryo, and the latter is obtained through a genetic reprogramming procedure in which terminally differentiated somatic cells are reversed back to the pluripotent state (*Fan et al., 2017*). Crucial morphological changes take place during PSCs differentiation. For example, highly dense PSCs colonies may produce a loosely organized cell structure. These morphological transformations can be rather obvious to the naked eye. However, they are subjective. Therefore, they are not a suitable measurement of cell differentiation.
In iPSC-derived cells, the cellular physiology can be observed specifically, making iPSCs useful for drug screening, disease analysis, and regenerative medicine. In addition, mature endothelial cells that have been differentiated through iPSCs (*Zhang et al., 2017*) can be used for disease modeling and organ formation. The cellular pathologies of Moyamoya disease, aortic valve calcification, and pulmonary arterial hypertension can also be improved by using iPSC-derived endothelial cells (iPSC-ECs) (*Theodoris et al., 2015*; *Hamauchi et al., 2016*; *Gu et al., 2017*).

Cellular reprogramming procedures have a significant relationship to morphological changes (*Butler & Wallingford, 2017*). Therefore, colony determination is one key problem limiting the quality and consistency of iPSCs and contributes to the isolation and purification of these colonies. A well-trained cell culture expert would be able to mitigate this issue, but would need the time and budget to do so. In addition, human recognition errors could contribute to misconceptions. In order to maintain a homogeneous culture of undifferentiated cells and downstream expansion, the quality of the iPSC colony identification must be controlled. A lack of the necessary downstream differentiation into functional cells is possible if the colony identification is not consistent. Therefore, an automated quantitative methodology with stable or constant colony maturation identification is needed to effectively aid biologists during the iPSCs production stages (*Fan et al., 2017*).

Application of CNNs in pluripotent stem cells studies was reported in a study conducted by *Kavitha et al. (2017)*. In this study, a vector-based convolutional neural network (V-CNN) was developed using the extracted features of the iPSC colony for distinguishing colony characteristics and was compared to SVM classifier using morphological, textural and combined features. The performance of the V-CNN model was examined using five-fold cross validation and it was found the precision, recall and F-measure values were much higher than the SVM by 87–93%. The V-CNN model also showed higher accuracy values for determining the quality of colonies based on morphological (95.5%), textural (91.0%) and combined features (93.2%) as compared to the SVM classifier (87.6, 83.3 and 83.4% respectively). Likewise, the accuracy of the feature sets was higher than 90% for the V-CNN model as compared to SVM that just yielded around 75–77%. This proved the V-CNN model performs better than the SVM classifier for iPSCs colony classification.

Then, *Fan et al. (2017)* reported on a time-lapse-based imaging study using bright-field microscopes that measured the morphological changes of the mesenchymal-to-epithelial transition during the cellular reprogramming that precedes iPSCs colony formation. However, the use of iPSCs in further applications may be limited due to the quality of iPSCs, which can only be checked by colony determination. Currently, colony determination is done by experienced cell culture experts and is time-consuming, expensive, error-prone, and inconsistent, which can lead to error and misjudgment. Therefore, an automated system for consistent colony determination is needed to help biology experts studying the iPSCs formation process. To this end, an ML model for the classification, segmentation, and statistical analysis of colony selection has been established. This model can spot changes in the cellular texture of reprogrammed human somatic cells as early as 7 days after the 20–24-day process. Furthermore, a scientific model was created to statistically

anticipate the best iPSC selection point after quantitatively resolving the reprogramming process and iPSC colony formation. These experiments in detection and prediction were verified biologically and evaluated with validation datasets. The findings of this study concluded that the colonies detected by the algorithm had non-significant differences in their biological features when compared to the colonies that were processed manually using typical molecular methods.

To identify endothelial cells produced from iPSCs, *Kusumoto et al. (2018)* utilized CNNs based on an automated methodology for this process without immunostaining or lineage tracing. In the study, 200 images were acquired from one of four independent tests. Of these, 640 were utilized for training, and 160 were utilized for validation. This study trained the networks to identify endothelial cells produced from iPSCs in phase-contrast images based only on the morphological features of the cells. This study was validated by contrasting the immunofluorescence staining of the CD31 marker for the endothelial cells. The efficiency of the method parameters was improved to make the prediction error-free. This improvement was done iteratively and automatically. The results of this study prove that the prediction accuracy of a CNN is directly dependent on the depth of the network and the pixel size of the images to be analyzed. The study also shows via K-fold cross-validation that CNNs can be improved to identify endothelial cells based on their morphological features alone.

In an investigation by *Waisman et al. (2019)*, light microscopic images of pluripotent stem cells were used to train a CNN to distinguish pluripotent stem cells from early differentiating cells. Images of the epiblast-like cells that were the result of mouse ESCs differentiating after induction were taken at different time intervals. It was found that the CNN model could be trained to recognize the ESCs from differentiating cells within 24–48 h with an outstanding accuracy higher than 99%. This trained CNN was found to detect differentiating cells only minutes after the cells were stimulated to differentiate. The CNN also performed well at mesoderm differentiation in human iPSCs. So far, this approach is the best cell assay for verifying differentiation in such a short timeframe, due to its accuracy (which is close to 100%) and low cost. The performance of the CNN in accurately identifying cellular morphology in microscopic images will definitely have substantial effects on the ways cell assays are conducted in the future. Table 2 summarizes the overall applications of CNNs in pluripotent stem cells studies since 2017.

## Future applications of artificial intelligence technology in hematopoietic stem cell analysis

It has been shown that AI technologies via CNNs offer a better platform for the study of iPSCs, particularly for the analysis of colonies. Thus, the applications of CNNs are likely to expand in the near future to other types of stem cell analysis. HSCs could also benefit from these advances in CNNs. In addition to iPSCs, HSCs are among the most established and widely utilized stem cell sources for medical and research applications. HSCs are vital to the maintenance of the hematopoietic system (*Butko, Pouget & Traver, 2016*). They are a valuable resource for stem cell–based therapy and have been used in various hematological research studies, ranging from toxicological, developmental biology, and

Ramakrishna et al. (2020), *PeerJ*, DOI 10.7717/peerj.10346

**Table 2  Applications of convolutional neural network on pluripotent stem cells studies since 2017.**

| Reference | Title | Objective | Methodology | Findings |
|---|---|---|---|---|
| *Kavitha et al (2017)* | Deep vector-based convolutional neural network approach for automatic recognition of colonies of induced pluripotent stem cells | A V-CNN model is developed to distinguish colony characteristics based on extracted features of the iPSC colony | • A transfer function from the feature vectors to the virtual image was generated at the front of the CNN in order for classification of feature vectors of healthy and unhealthy colonies | • Precision, recall, and F-measure values of the V-CNN model were higher than the SVM classifier with a range of 87–93% |
| | | | • The performance of V-CNN model in distinguishing colonies was compared with the competitive SVM classifier based on morphological, textural, and combined features | • For the quality of colonies, the V-CNN model showed higher accuracy values based on morphological (95.5%), textural (91.0%), and combined (93.2%) features than those estimated with the SVM classifier (86.7, 83.3, and 83.4%, respectively) |
| | | | • Five-fold cross-validation was used to investigate the performance of the V-CNN model | • The accuracy of the feature sets using five-fold cross-validation was higher than 90% for the V-CNN model as compared to SVM that just yielded around 75–77% |
| *Fan et al (2017)* | A Machine Learning Assisted, Label-free, Non-invasive Approach for Somatic Reprogramming in Induced Pluripotent Stem Cell Colony Formation Detection and Prediction | A computer vision system to recognize and assist the classification of induced pluripotent mouse embryonic stem cells colonies from microscopic images | • CNN is used as classifier to recognize colonies | • This algorithm shows no significant differences (Pearson coefficient $r > 0.9$) in detection and prediction of colonies in terms of biological features compared to manually processed colonies |
| | | | • Colonies are located and their boundaries are detected by a semi-supervised segmentation method | • Evaluation was completed by standard immunofluorescence staining, quantitative polymerase chain reaction (QPCR), and RNA-Seq for verification of pluripotency |
| | | | • Growth phase and maturation time window of colony formation was estimated with trained Hidden Markov Model (HMM) during the reprogramming procedure | |
| | | | • This system can predict the best selection time window for iPSC colonies to prevent random differentiation caused by overgrowth using data from colonies traced via time-lapse | |

Ramakrishna et al. (2020), *PeerJ*, DOI 10.7717/peerj.10346

**Table 2** (*continued*)

| Reference | Title | Objective | Methodology | Findings |
|---|---|---|---|---|
| *Kusumoto et al (2018)* | The application of CNN to stem cell biology | Recognition of iPSC-derived endothelial cells based on morphological features using CNN | ● Images were collected at day 6 of differentiation | ● The results proved that identification of iPSC-derived endothelial cells can be made based on morphology alone |
| | | | ● 200 images were attained from each of four independent experiments and of these, 640 images were allocated for training and 160 were used for validation | |
| | | | ● Under K-fold validation, three experiments that rendered 600 images were used for training phase and 200 images from one experiment were allocated for validation, in every possible combination | |
| | | | ● CNN based LeNet and AlexNet model used | |
| | | | ● F1 scores determined the performance, which indicated the aggregate of recall and precision, and on accuracy (portion of true predictions) | |
| *Waisman et al (2019)* | Deep Learning Neural Networks Highly Predict Very Early Onset of Pluripotent Stem Cell Differentiation | Use CNNs to precisely forecast the beginning of PSC differentiation in transmitted light microscopy images | ● Mouse ESCs were cultured in distinct conditions to maintain the ground state of pluripotency | ● CNN model could be trained to recognize the ESCs from differentiating cells within 24–48 h with an outstanding accuracy higher than 99%. |
| | | | ● Images were taken randomly through EVOS microscope at consecutive hours post differentiation | ● This trained CNN was found to detect differentiating cells only minutes after the cells were stimulated to differentiate |
| | | | ● Light microscopic images of pluripotent stem cells were used to train a CNN to distinguish pluripotent stem cells from early differentiating cells | ● So far, this approach is the best cell assay for verifying differentiation in such a short timeframe, due to its accuracy (which is close to 100%) and low cost |
| | | | ● CNN based ResNet and DenseNet model used | ● The performance of the CNN in accurately identifying cellular morphology in microscopic images will definitely have substantial effects on the ways cell assays are conducted in the future. |
| | | | ● 2134 images were selected for training and 400 for validation (200 in each group). Independent prediction after training phase was done with 100 images (50 per group) | |

**Notes.**

CNN, convolutional neural network; iPSC, induced pluripotent stem cells; ESC, embryonic stem cell; SVM, support vector machine; V-CNN, vector-based convolutional neural network.

drug testing applications (*Ng & Alexander, 2017*). Therefore, it is crucial to study and verify the properties of HSCs using HSC-based functional assays. One of the fundamental HSC-based functional assays for HSCs is the colony-forming unit (CFU) assay. The CFU assay is a widely used assay to measure the proliferation and differentiation abilities of individual hematopoietic stem and progenitor cells (HSPCs) within a sample. The measurements are carried out by observing the colonies produced by each input progenitor cell for each respective myeloid, lymphoid, or erythroid lineage. About 7–14 days of cultures are required to allow the colonies to grow to a size that allows for accurate counting and identification (*Chow et al., 2018*). However, accurately identifying the different colony types in these cultures is challenging, often difficult to achieve, time-consuming, and error-prone, particularly for those who are inexperienced or are using atypical cell sources or samples. Thus, staff training is often an essential part of counting and classifying the colonies in a CFU assay. One way to simplify the analysis of CFU assays and to overcome the limitations of manual analysis is to perform colony counting and classification automatically without compromising the information on lineage-specific progenitor cell growth.

As discussed earlier in this review, AI has shown a promising ability to extract features and learning patterns from complex data and images. However, no study has reported on the use of AI to perform automatic colony counting or to classify HSC-derived CFUs based on simple images taken in a transmission light microscope. Thus, we believe that AI technologies, particularly via CNNs, can be utilized to process images for automatic cell pattern recognition and morphological analyses of CFU colonies cultivated from hematopoietic progenitors of various lineages comprised of myeloid, lymphoid, and erythroid lineages. As illustrated in Fig. 4, these progenitors are morphologically different and require skilled and trained staff to be analyzed accurately (*Chow et al., 2015*). Due to the significant limitations associated with manual analysis, attempts to more extensively use hematopoietic CFU assays in clinical and research settings will be hampered. Thus, automated systems using AI could be a potential solution to overcome these hurdles and impact the future of stem cell studies in science and medicine.

## CONCLUSIONS

There is compelling evidence for the remarkable potential of AI in the fields of medical and health sciences. AI is expected to accelerate progress in biomedical research and to have diagnostic, pharmaceutical, and therapeutic applications in multiple healthcare sectors. Recently, convolutional neural networks (CNNs), a subfield of AI, have been applied to analyze medical images due to their outstanding abilities in image classification, detection, and segmentation. For stem cell studies, CNNs have revolutionized the morphological analysis of stem cells. In particular, CNNs can identify induced pluripotent stem cells (iPSCs) with high accuracy using microscopic images that replace the conventional detection methods using molecular labeling techniques. This breakthrough significantly advances the application of iPSCs in stem cell–based therapy and research. Because CNNs are able to distinguish cells with high accuracy based on morphologic and textural changes, we believe that CNNs will create a future field where AI is used for the analysis of other

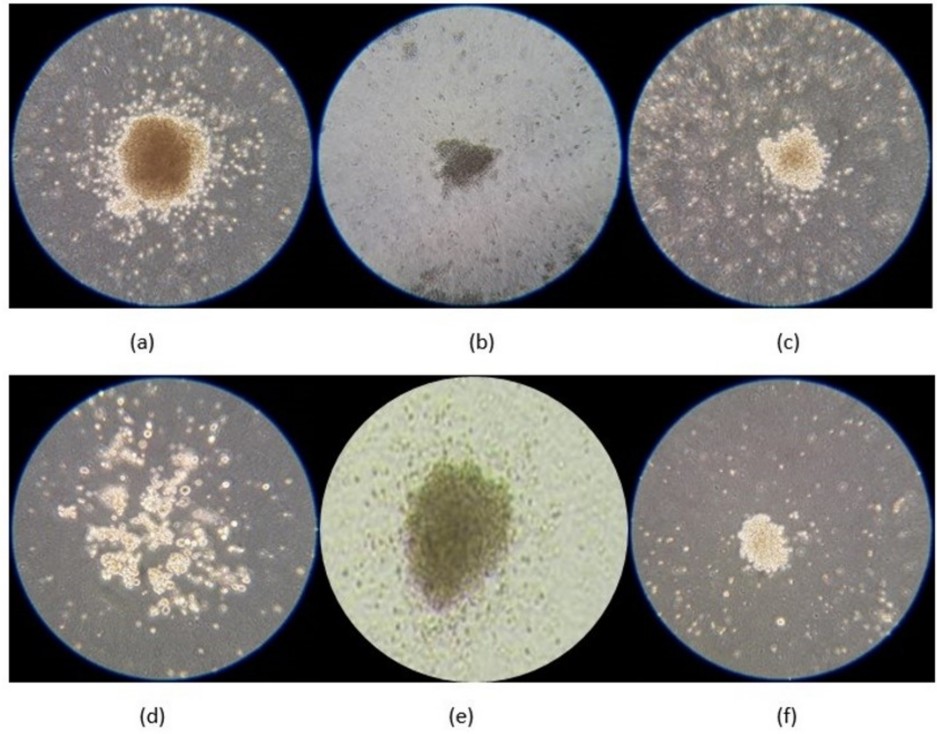

**Figure 4   Microscopic images of lineage-committed hematopoietic progenitors using Colony Forming Unit assay as taken using a smartphone (Samsung Galaxy J6).** Myeloid progenitors comprise of: (A) colony forming unit (CFU) for granulocyte-macrophage (CFU-GM), (B) granulocyte (CFU-G) and (C) macrophage (CFU-M). Erythroid progenitors comprise of; (D) CFU for erythroid (CFU-E) and (E) burst-forming unit erythroid (BFU-E). (F) Lymphoid progenitor for pre-B lymphoid.

stem cells, including hematopoietic stem cells (HSCs). In addition, CNNs can be used for automatic cell pattern recognition and morphological analysis to distinguish colonies from various hematopoietic progenitors in myeloid, lymphoid, and erythroid lineages. Furthermore, cloud-based technologies can be utilized in the analysis of HSC colonies from different hematopoietic lineages by capturing microscopic images on smartphones. In conclusion, AI-based technologies are likely to have a significant impact on accurate cellular morphology recognition and stem cell studies in the near future.

## ACKNOWLEDGEMENTS

The authors would like to thank the Biotechnology Laboratory, Faculty of Health Sciences, UKM for providing facilities to conduct this research.

### Funding
This study was funded by GERAN UNIVERSITI PENYELIDIKAN (GUP-2019-055). The funders had no role in study design, data collection and analysis, decision to publish, or preparation of the manuscript.

### Grant Disclosures
The following grant information was disclosed by the authors:
GERAN UNIVERSITI PENYELIDIKAN: GUP-2019-055.

### Competing Interests
The authors declare there are no competing interests.

### Author Contributions
- Ramanaesh Rao Ramakrishna conceived and designed the experiments, performed the experiments, analyzed the data, prepared figures and/or tables, authored or reviewed drafts of the paper, and approved the final draft.
- Zariyantey Abd Hamid, Wan Mimi Diyana Wan Zaki and Aqilah Baseri Huddin conceived and designed the experiments, performed the experiments, analyzed the data, authored or reviewed drafts of the paper, and approved the final draft.
- Ramya Mathialagan performed the experiments, authored or reviewed drafts of the paper, and approved the final draft.

### Data Availability
This is a literature review article; no experimental data was collected.

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
