# Peer review of "Stem cell imaging through convolutional neural networks: current issues and future directions in artificial intelligence technology"

_PeerJ, doi:10.7717/peerj.10346_

## Round 0.1 · original submission · Major Revisions

Your manuscript has now been seen by 2 reviewers. You will see from their comments below that while they find your work of interest, important points are raised. We are interested in the possibility of publishing your review, but would like to consider your response to these concerns in the form of a revised manuscript before we make a final decision on publication. We therefore invite you to revise and resubmit your manuscript, taking into account the points raised. Please use clear, unambiguous, technically and grammatically correct English throughout. The presentation should also be checked by a fluent English-speaker. Please highlight all changes in the manuscript text file.

Reviewer 1 ·

Basic reporting

Review of Manuscript: #49552 entitled ‘Stem Cells imaging through Artificial Intelligence
Technology: Current issues and future directions’ by Ramakrishna and colleagues.

Dear The Editor

The above manuscript by Ramakrishna and colleagues is a review manuscript describing the role of artificial intelligence technology in stem cell imaging. The discovery of stem cells opened many doors in medicine in terms of treatment of many diseases and conditions with no cure at present. The use of stem cells in medicine is however hampered by the great heterogeneity observed and also their functional study. Whilst great work has been done in stem cell research more is required before they reach their full potential in their application in medicine. This review manuscript argues that artificial intelligence can be used to study stem cells better than currently is. For example, development of automated instrument for analysis of iPSC colony is fundamental to the use of stem cells. This review manuscript is an attempt by the authors to present an overview of artificial intelligence and its role in biomedical field and associated stem cells-based therapy and research.

Overall the manuscript requires major changes before it is ready for publication. Most importantly, a native English speaking person must edit the manuscript before re-submission as there are many errors, including simple mistakes such as singular and plural mistakes.

Specific Points
1. Whilst the Abstract mention that the manuscript will discuss the use of artificial intelligence in stem cell-based therapy, the manuscript contains minimal information on stem cell based therapy and the use of AI in this field.
2. Lines 42-43. The definition of stem cell given is wrong. As given, it implies a cell can become terminally differentiated whilst at the same time it can self-renew. This is wrong. A stem cell can divide into two daughter cells with one cell undergoing terminal differentiation whilst the other one maintains self-renewal properties.
3. Lines 113 -132. The selection of articles and other materials used in the construction of the manuscript must be given in a Figure form. Add inclusion and exclusion criteria to the Figure as well.
4. Lines 162 onwards. Sections for Machine learning and Deep learning require more than one Figure. The authors must illustrate what they are describing in these sections.
5. Lines 247- Sections on ‘CNN and Medical Analysis’ and ‘CNN in Cell Biology’ is missing key publications. The following references must be added to these sections.
PMID: 32416778
PMID: 32213531
PMID: 32061286
PMID: 31367025
PMID: 30617339
PMID: 31592719
PMID: 31313972
PMID: 32602650

Experimental design

Review of Manuscript: #49552 entitled ‘Stem Cells imaging through Artificial Intelligence
Technology: Current issues and future directions’ by Ramakrishna and colleagues.

Dear The Editor

The above manuscript by Ramakrishna and colleagues is a review manuscript describing the role of artificial intelligence technology in stem cell imaging. The discovery of stem cells opened many doors in medicine in terms of treatment of many diseases and conditions with no cure at present. The use of stem cells in medicine is however hampered by the great heterogeneity observed and also their functional study. Whilst great work has been done in stem cell research more is required before they reach their full potential in their application in medicine. This review manuscript argues that artificial intelligence can be used to study stem cells better than currently is. For example, development of automated instrument for analysis of iPSC colony is fundamental to the use of stem cells. This review manuscript is an attempt by the authors to present an overview of artificial intelligence and its role in biomedical field and associated stem cells-based therapy and research.

Overall the manuscript requires major changes before it is ready for publication. Most importantly, a native English speaking person must edit the manuscript before re-submission as there are many errors, including simple mistakes such as singular and plural mistakes.

Specific Points
1. Whilst the Abstract mention that the manuscript will discuss the use of artificial intelligence in stem cell-based therapy, the manuscript contains minimal information on stem cell based therapy and the use of AI in this field.
2. Lines 42-43. The definition of stem cell given is wrong. As given, it implies a cell can become terminally differentiated whilst at the same time it can self-renew. This is wrong. A stem cell can divide into two daughter cells with one cell undergoing terminal differentiation whilst the other one maintains self-renewal properties.
3. Lines 113 -132. The selection of articles and other materials used in the construction of the manuscript must be given in a Figure form. Add inclusion and exclusion criteria to the Figure as well.
4. Lines 162 onwards. Sections for Machine learning and Deep learning require more than one Figure. The authors must illustrate what they are describing in these sections.
5. Lines 247- Sections on ‘CNN and Medical Analysis’ and ‘CNN in Cell Biology’ is missing key publications. The following references must be added to these sections.
PMID: 32416778
PMID: 32213531
PMID: 32061286
PMID: 31367025
PMID: 30617339
PMID: 31592719
PMID: 31313972
PMID: 32602650

Validity of the findings

No comment

Additional comments

Review of Manuscript: #49552 entitled ‘Stem Cells imaging through Artificial Intelligence
Technology: Current issues and future directions’ by Ramakrishna and colleagues.

Dear The Editor

The above manuscript by Ramakrishna and colleagues is a review manuscript describing the role of artificial intelligence technology in stem cell imaging. The discovery of stem cells opened many doors in medicine in terms of treatment of many diseases and conditions with no cure at present. The use of stem cells in medicine is however hampered by the great heterogeneity observed and also their functional study. Whilst great work has been done in stem cell research more is required before they reach their full potential in their application in medicine. This review manuscript argues that artificial intelligence can be used to study stem cells better than currently is. For example, development of automated instrument for analysis of iPSC colony is fundamental to the use of stem cells. This review manuscript is an attempt by the authors to present an overview of artificial intelligence and its role in biomedical field and associated stem cells-based therapy and research.

Overall the manuscript requires major changes before it is ready for publication. Most importantly, a native English speaking person must edit the manuscript before re-submission as there are many errors, including simple mistakes such as singular and plural mistakes.

Specific Points
1. Whilst the Abstract mention that the manuscript will discuss the use of artificial intelligence in stem cell-based therapy, the manuscript contains minimal information on stem cell based therapy and the use of AI in this field.
2. Lines 42-43. The definition of stem cell given is wrong. As given, it implies a cell can become terminally differentiated whilst at the same time it can self-renew. This is wrong. A stem cell can divide into two daughter cells with one cell undergoing terminal differentiation whilst the other one maintains self-renewal properties.
3. Lines 113 -132. The selection of articles and other materials used in the construction of the manuscript must be given in a Figure form. Add inclusion and exclusion criteria to the Figure as well.
4. Lines 162 onwards. Sections for Machine learning and Deep learning require more than one Figure. The authors must illustrate what they are describing in these sections.
5. Lines 247- Sections on ‘CNN and Medical Analysis’ and ‘CNN in Cell Biology’ is missing key publications. The following references must be added to these sections.
PMID: 32416778
PMID: 32213531
PMID: 32061286
PMID: 31367025
PMID: 30617339
PMID: 31592719
PMID: 31313972
PMID: 32602650

Annotated reviews are not available for download in order to protect the identity of reviewers who chose to remain anonymous.

·

Basic reporting

Ramakrishna et al reviewed the potential applications of AI in the stem cell field. This review manuscripts aims to go over the literature that focused two broad fields, although it is hard to tell real differences between applications of AI in other cell types. Anyhow, this review is novel, and there was an intention to write it as a systematic review, and it is presented in that way. However, it lacks the solid criteria needed to consider as one of them, and hence this format is misleading. Moreover, it is unclear why this review was written. Although I perfectly understand the potential uses of AI in the stem cell field, this is not clearly stated in this review. It is definitively vague in this matters.

Experimental design

As I said, the review is presented as a systematic review, and the authors did an excellent job in collecting the few papers in the field. However, information is not presented as the interested reader would expect. Overall, it is not clear what information a stem cell researcher would be able to take out from a review like this. For example, figure 1 is correct. It shows a diagram of a CNN, but there is no legend that extensively explains what this is about, nor it is explained in the text. It is really uninformative in this way.

Validity of the findings

Overall, the review lacks deepness. It is a shallow review of what Ai can do in stem cells. Many times, the text does not flow into any conclusion, and frequently the reader is taken from one subject to another without any reasonable connection. A few examples are lines 171 (no explanation what are these three fields), lines 184 to 187 (what these accomplishment really mean), the section CNN in medical analysis (it is non informative for this review), etc. Moreover, most interesting details for a stem cell biologist are not fully explained. For example, the whole paragraph from line 297 to 310. Perhaps this is one of the most important paragraph in this manuscript, but unfortunately does not describes in details the main findings of these papers, as well as makes some comments about U-Nets, which only adds confusion. I appreciate that the authors also cited our work, which I don't think it was fully understood. We showed an extremely example of detecting stem cell differentiation, but most interesting, incredible fast and cheap. These findings are not discussed and we believe that they are of paramount importance for stem cell biologists. Finally, the field of stem cells is too wide, from EC to haematopoietic stem cells. However, they are extremely different, from biology to medical applications. I think that this distinction should be clear.

Additional comments

Unfortunately, I believe the authors did not fill the expectations of a potential reader for this manuscript. I think that there is a strong potential for this review, but it essentially needs to be rewritten. Some major questions for the stem cell field would remain without answers if this manuscript is published in this format.

---

## Round 0.2 · accepted · Accept

Thank you for the revised manuscript and response letter. I am pleased to inform you that your manuscript has been accepted for publication.

Reviewer 1 ·

Basic reporting

The manuscript reads very well now. After going through several times, English language is good for publication. References have been added and is adequate.

Experimental design

The structure of the Review has been adjusted and is good. A Figure has been added to the Survey Methodology, making it clear and understandable to readers.

Validity of the findings

Conclusions drawn by the authors appear plausible and applicable. The authors did a good job of addressing the issues l raised in the previous review.

Additional comments

The review is good for publication now